# Head-to-Head Comparison of Nasopharyngeal, Oropharyngeal and Nasal Swabs for SARS-CoV-2 Molecular Testing

**DOI:** 10.3390/diagnostics13020283

**Published:** 2023-01-12

**Authors:** Kasper Daugaard Larsen, Mads Mose Jensen, Anne-Sophie Homøe, Elisabeth Arndal, Grethe Badsberg Samuelsen, Anders Koch, Xiaohui Chen Nielsen, Preben Homøe, Tobias Todsen

**Affiliations:** 1Department of Otorhinolaryngology and Maxillofacial Surgery, Zealand University Hospital, 4600 Koege, Denmark; 2Department of Otolaryngology—Head and Neck Surgery and Audiology, Rigshospitalet, 2100 Copenhagen, Denmark; 3Department of Otorhinolaryngology, Nordsjaellands Hospital, 3400 Hilleroed, Denmark; 4Department of Infectious Diseases, Rigshospitalet, 2100 Copenhagen, Denmark; 5Staten Serum Institut (SSI), 2100 Copenhagen, Denmark; 6Department of Clinical Microbiology, Zealand University Hospital, 4600 Koege, Denmark; 7Department of Clinical Medicine, University of Copenhagen, 1353 Copenhagen, Denmark

**Keywords:** COVID-19, SARS-CoV-2, diagnostic testing, swab, molecular diagnostic techniques

## Abstract

Nasopharyngeal swabs (NPS) are considered the gold standard for SARS-CoV-2 testing but are technically challenging to perform and associated with discomfort. Alternative specimens for viral testing, such as oropharyngeal swabs (OPS) and nasal swabs, may be preferable, but strong evidence regarding their diagnostic sensitivity for SARS-CoV-2 testing is still missing. We conducted a head-to-head prospective study to compare the sensitivity of NPS, OPS and nasal swabs specimens for SARS-CoV-2 molecular testing. Adults with an initial positive SARS-CoV-2 test were invited to participate. All participants had OPS, NPS and nasal swab performed by an otorhinolaryngologist. We included 51 confirmed SARS-CoV-2-positive participants in the study. The sensitivity was highest for OPS at 94.1% (95% CI, 87 to 100%) compared to NPS at 92.5% (95% CI, 85 to 99%) (*p* = 1.00) and lowest for nasal swabs at 82.4% (95% CI, 72 to 93%) (*p* = 0.07). Combined OPS/NPS was detected in 100% of cases, while the combined OPS/nasal swab increased the sensitivity significantly to 96.1% (95% CI, 90 to 100%) compared to that of the nasal swab alone (*p* = 0.03). The mean Ct value for NPS was 24.98 compared to 26.63 for OPS (*p* = 0.084) and 30.60 for nasal swab (*p* = 0.002). OPS achieved a sensitivity comparable to NPS and should be considered an equivalent alternative for SARS-CoV-2 testing.

## 1. Introduction

A high quality upper respiratory tract specimen is the most important step in the diagnosis of COVID-19 [1]. Molecular testing, such as real-time reverse transcription–polymerase chain reaction (RT-PCR), is considered the test with the highest sensitivity to detect SARS-CoV-2 (severe acute respiratory syndrome coronavirus 2) but is highly depending on a representative respiratory tract sample [2]. A nasopharyngeal swab (NPS) is considered to have the highest diagnostic yield and is recommended as the gold standard for SARS-CoV-2 testing [3,4]. However, NPS is also technically difficult to perform, associated with pain and discomfort for the patients and exposing healthcare workers to an infectious risk [5]. Alternative swabs for viral testing, such as oropharyngeal swabs (OPS) and nasal swabs, are therefore preferred for reference sample in some countries due to lower discomfort and better possibilities for mass testing [6,7]. Earlier studies during the COVID-19 pandemic have questioned the diagnostic sensitivity of especially OPS compared to NPS [3,8,9]. However, many of the previous studies were either retrospective or lacking descriptions of sample procedures and the training process of the health care workers [10]. Still, there is a need to provide better evidence with head-to-head comparison studies collecting different high-quality and standardized upper respiratory specimen types.

The aim of this prospective diagnostic study was to head-to-head compare the sensitivity of OPS, NPS and nasal swab specimens collected as paired samples in each participant for SARS-CoV-2 molecular testing.

## 2. Methods

We conducted a prospective clinical trial comparing NPS, OPS and nasal swab for SARS-CoV-2 molecular detection. This study was part of another clinical trial exploring the olfactory and gustatory dysfunction among Danish citizens infected with SARS-CoV-2 [11]. Written and verbal informed consent was obtained from all participants and the protocol was approved by the Danish Regional Ethics Committee (protocol number: SJ-714 and Letter ID 4336732).

### 2.1. Participants

Adults (18 years or above) with an initial positive SARS-CoV-2 test from COVID-19 outpatient test facilities in the Capital Region and Region Zealand, Denmark were invited to participate in the study. Participants were enrolled if they had a positive SARS-CoV-2 test of less than 10 days old and agreed to participate in the study with informed consent. Via a QR-code, participants answered a questionnaire about symptoms (fever, cough, loss or change in smell or taste, sore throat, blocked or runny nose, being sick) available on a Research Electronic Data Capture (REDCap, Vanderbilt, TN, USA) database.

Participants with a positive SARS-CoV-2 molecular test were contacted and invited for collection of respiratory specimens.

### 2.2. Procedures for Collecting Upper Respiratory Specimens

All participants were examined at the Departments of Otorhinolaryngology at either Zealand University Hospital, Nordsjaellands Hospital or Rigshospitalet where special facilities were used to accommodate the risk of SARS-CoV-2 transmission. In addition to the collection of upper respiratory tract specimens, the participants also had their olfactory and gustatory function assessed, reported in another publication [11]. A consultant or a registrar in otorhinolaryngology performed an OPS, an NPS, and a nasal swab, respectively. Each specimen was placed into separate sterile tubes with 2 mL of transport medium (Meditec A/S, Roskilde, Denmark) and sent for RT-PCR analysis at the Departments of Clinical Microbiology at Zealand University Hospital, Herlev Hospitalet or Rigshospitalet.

The NPS was collected using a flexible minitip flocked swab (COPAN diagnostics Inc, Italy), while the OPS and nasal swab were collected using a rigid-shaft flocked swab (Meditec A/S, Denmark). The procedures for collecting upper respiratory tract specimens followed the description below and are illustrated in Figure 1.

The NPS were performed with the patient’s head tilted slightly back. The swab was inserted into the nasal cavity and pointed towards the earlobe of the patient following the floor of the nose.

The swab was inserted approximately 8–11 cm deep until resistance was met in the form of the posterior wall of the nasopharynx [12]. The swab was left for a couple of seconds in the nasopharynx before it was rotated about three times and withdrawn. If the swab met noteworthy resistance before the nasopharynx was reached, the swab was withdrawn and the insertion direction changed, or the opposite nasal cavity was used.

The OPS was performed with use of a tongue depressor to improve visualization. The specimen was collected from both palatine tonsils and the posterior oropharyngeal wall with a painting and rotating movement of the swab without touching the cheeks, the teeth, or the gums [13].

The nasal swab was conducted in the same way as the NPS, but the swab was only inserted approximately 1–3 cm into the nasal cavity and brushed along the septum and the inferior nasal concha and rotated about three times before withdrawal [1].

### 2.3. Laboratory Methods

All samples were stored at 2–6 °C at the test site before transportation to the department for Clinical Microbiology either at Zealand University Hospital, Herlev Hospital or Rigshospitalet depending on the site of inclusion and testsing.

All samples from Zealand University hospital were tested with Allplex^TM^ SARS-CoV-2 real-time PCR assay (Seegene, Seoul, South Korea). The system is automated using the STARlet (Seegene, South Korea) for extraction of RNA and PCR setup, and Seegene Viewer for analyzing the data. The Allplex^TM^ SARS-CoV-2 assay includes internal control (IC) and four targets: E gene, N gene, RdRP gene and S gene. RdRP and S genes use the same fluorophore and cannot be distinguished. The assay has a software algorithm with cycle threshold (Ct) cut-off values ≤ 40 for IC and all four targets. Samples that are only positive for E gene are scored as inconclusive by the software, because the E gene is not SARS-CoV-2 specific, but a pan-Sarbecovirus target.

The respiratory specimens collected at Herlev Hospital or Rigshospitalet were analyzed on the RT-PCR systems available for the qualitative detection of SARS-CoV-2 RNA at the sample time, and therefore many different systems were used (See Appendix A). However, all samples collected from a participant were tested with the same RT-PCT assay.

### 2.4. Outcome Measures

The main objective of this study was to compare the sensitivity of RT-PCR detection of SARS-CoV-2 in OPS, NPS, and nasal swabs collected in identified positive individuals. A study participant was defined as having a present SARS-CoV-2 infection, if they had a prior confirmed positive test and one or more respiratory specimens were RT-PCR positive in the study (gold standard) [14]. Participants with prior positive test but who produced negative tests in all three samples (OPS, NPS, and nasal swab) collected as a part of the study were excluded from the analysis. The RT-PCR results from the OPS, NPS and nasal swab were used to compare the SARS-CoV-2 detection rate. The sensitivity of the combined OPS/NPS and OPS/nasal swab specimens were defined as positive if both or one of the combined specimens were positive. Further, the mean Ct values for the N gene segment from SARS-CoV-2 RNA RT-PCR were calculated for each type of specimen in 24 participants analyzed at Zealand University Hospital with the same SARS-CoV-2 RT-PCR assay. The Ct values for the remaining participants were not included in the analysis because the specimens were analyzed on different RT-PCR systems and therefore not comparable.

### 2.5. Statistical Methods

A McNemar test was used to compare the differences in test sensitivity between OPS, NPS, nasal swab and the combined samples. Ct values were compared using the Wilcoxon matched pairs signed-rank test. A 5% significance level was applied. Statistical analyses and graphics were performed using IBM SPSS (Statistics for Windows, Version 28.0. Armonk, NY, USA: IBM Corp) and RStudio: Integrated Development for R. RStudio, PBC, Boston, MA, USA.

## 3. Results

From June 2020 to May 2021, 56 participants with a positive molecular test for SARS-CoV-2 were included in the study. Five participants were excluded from analysis due to negative RT-PCR test results in all three samples. In total, 51 participants were included in the analysis, 24, 20, and 7 participants were tested at Zealand University Hospital, Rigshospitalet, and Nordsjaellands Hospital, respectively. Out of the 51 participants included, 26 (51%) were female with the mean age of 42.8 years (range 18–81 years). Thirty-one (60.8%) participants were symptomatic at inclusion and the mean time from initial positive test to study inclusion was 4.9 days (range 1–10).

The number of positive RT-PCR test results for the different sample methods were 48, 47 and 42 for OPS, NPS and nasal swabs, respectively (See Table 1). The diagnostic sensitivity for the different sample methods was 94.1% (95% Confidence Interval (CI), 87 to 100%) for OPS, 92.5% (95% CI, 85 to 99%) for NPS and 82.4% (95% CI, 72 to 93%) for nasal swabs. For the combined samples, the diagnostic sensitivity was 100% for OPS/NPS specimens and 96.1% (95% CI, 90 to 1.00%) for the combined OPS/nasal swab specimens. The sensitivity was 1.6% higher for OPSs than for NPSs (*p* = 1), 11.7% higher for OPSs than for nasal swabs (*p* = 0.070) and 9.8% higher for NPSs than for nasal swabs (*p* = 0.063), see Table 2. Combined OPS/NPS specimens had a 5.8% (*p* = 0.25), a 7.8% (*p* = 0.125) and a 17.6% (*p* = 0.004) higher sensitivity than OPS, NPS and nasal swabs, respectively.

The mean Ct value for NPS was 24.98 (4.24 SD) compared to 26.63 (4.60 SD) for OPS (*p* = 0.084) and 30.60 (4.75 SD) for nasal swab (*p* = 0.002) in 24 participants, see Figure 2. The mean Ct value for OPS was significantly lower than that of nasal swab (*p* = 0.005).

The central bar in the box plot represents the mean, the box represents the interquartile range, and the whiskers represent the range. Wilcoxon signed-rank tests (paired) were used to compare the Ct values.

## 4. Discussion

In this prospective clinical trial, we conducted a head-to-head comparison of three upper respiratory sampling methods for SARS-CoV-2 molecular testing. Our results support the use of OPSs as a comparable alternative to NPSs for SARS-CoV-2 testing. In contrast, nasal swabs had a lower sensitivity and significantly higher Ct values compared to OPSs and NPSs. The highest sensitivity was achieved by the collection of combined OPS/NPS or OPS/nasal specimen for SARS-CoV-2 molecular testing.

A strength in our study is the prospective design with use of standardized sample materials and molecular tests to ensure highly sensitive head-to-head comparison of SARS-CoV-2 detection between different sample methods. Further, we followed the recommendations from WHO and CDC for collecting upper respiratory tract specimens and ensured high quality swabbing were performed by otorhinolaryngologists. A limitation of the study is that participants included were selected based on a prior SARS-CoV-2 positive test (up to 10 days old) and may have been potentially tested in a later infectious stage. However, the selected design also ensured that all participants in the analysis had at least two independent positive RT-PCR tests, which therefore decreased the risk of false positive results. Besides the 24 participants who were tested with the same SARS-CoV-2 RT-PCR assay at Zealand University Hospital, the remaining 27 participants from Rigshospitalet and Herlev Hospital were tested on many different RT-PCR systems. The different methods used for molecular SARS-CoV-2 detection is a limitation of this study, and future studies should ensure all specimens are tested on the same RT-PCR system.

The SARS-CoV-2 virus accumulates mutations over time, resulting in genetic variation in the population of viral strains. The impact of virus genomic variants in specimen collection on sensitivity is unknown. The molecular tests used in this study are designed to detect multiple SARS-CoV-2 genetic targets, therefore less susceptible to the effect of genetic variation [15,16].

Another limitation of this study is the relatively small sample size, which may have resulted in underpowered statistical analyses.

NPS has been considered the highest-yield upper respiratory specimen for SARS-CoV-2 diagnostic testing during the COVID-19 pandemic [3]. It has therefore been the preferred sample for SARS-CoV-2 molecular testing globally. A review by Tsang et al. determined OPS to be inferior to NPS and nasal swab, and The Infectious Diseases Society of America advise against using OPS for SARS-CoV-2 diagnostic testing [17,18]. Although OPS is controversial in the scientific literature, China and some Northern European countries still use OPS as the preferred sampling method during mass testing in society [6,7]. Our findings indicate that OPS had comparable diagnostic sensitivity to NPS, which is surprising according to the current SARS-CoV-2 testing practice. However, previous systematic reviews report a significant variance in estimated OPS sensitivity between 52 and 100% [3,18]. The OPS procedure requires proper skills training to ensure correct sampling from the posterior oropharyngeal wall and avoid false negative results [19]. Previous studies exploring OPS for SARS-CoV-2 molecular testing are mostly retrospective and therefore lack detailed descriptions of the swabbing procedure, making it difficult to compare with our study. A reason for the improved OPS sensitivity in our study could be our swabbing technique, which included the use of tongue depressor and specimen collection from both palatine tonsils and the posterior wall of the oropharynx [20]. The training of the healthcare workers performing the OPS procedure may also have been insufficient in previous studies, while we have a more standardized and high-quality sampling technique performed by otorhinolaryngologists in our study [10,21,22]. This is also supported by another recent prospective study that determined OPS specimens to have comparable sensitivity with NPS specimens for SARS-CoV-2 detection [23].

Our findings can directly be used by clinicians and health care authorities to optimize testing strategies during the COVID-19 pandemic. The choice of respiratory samples has a direct influence on the sensitivity for the molecular test strategy, and this study suggests that OPS is comparable to NPS and can therefore be used as an alternative sample in an outpatient setting. The nasal swab had the lowest sensitivity (82%) and should therefore be used in combination with OPS to decrease the number of false negative test results. The combined OPS/nasal swab is also recommended by the National Health Service (NHS) in England and can be self-collected [24,25]. In contrast, the CDC recommend single OPS, NPS or nasal swabs as sufficient for SARS-CoV-2 testing and do not differentiate between sensitivity of specimen type. Our results determined that the highest sensitivity was obtained by the combined OPS/NPS specimen as recommended by WHO, but the sample technique would also be expected to provide most procedural discomfort, which may not be ideal for outpatient testing. However, the tradeoff between test sensitivity and cost/discomfort should always be considered based on the false negativity test consequences with possible transmission among vulnerable persons at hospitals and nursing homes [26]. This study demonstrates the importance of choosing the optimal sample strategy for SARS-CoV-2 molecular testing depending on the will to obtain the highest sensitivity possible. Future studies should therefore explore whether our findings can also be generalizable to SARS-CoV-2 community testing, SARS-CoV-2 antigen testing and testing for other upper respiratory tract specimens such as influenza viruses.

## 5. Conclusions

In conclusion, we determined that OPS has a sensitivity comparable to NPS specimens and should be considered as an equivalent alternative for SARS-CoV-2 testing. The nasal swab had the lowest sensitivity, while the combined OPS/NPS had the significantly highest sensitivity.

## Figures and Tables

**Figure 1 diagnostics-13-00283-f001:**
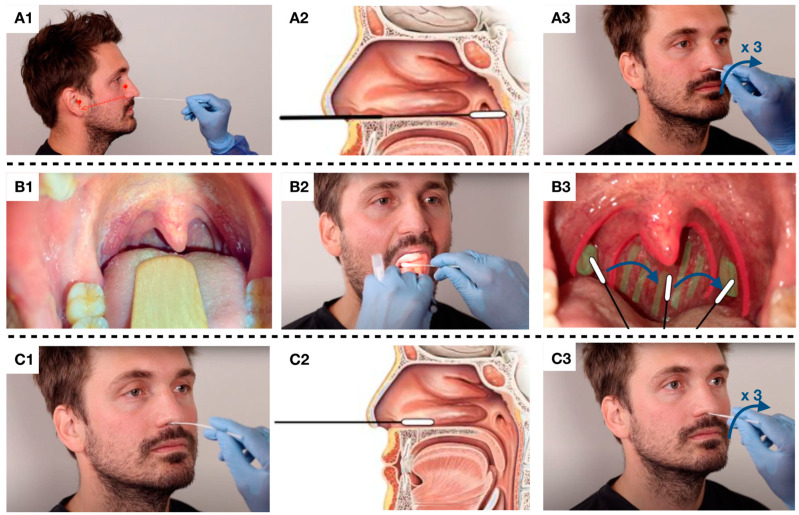
(**A1**): The nasopharyngeal swab was performed with the patient’s head tilted slightly back. The swab was inserted into the nasal cavity and pointed towards the earlobe. (**A2**): The swab was inserted approximately 8–11 cm deep until resistance was met in the form of the posterior wall of the nasopharynx. (**A3**): The swab was rotated three times and withdrawn. (**B1**,**B2**): The oropharyngeal swab was performed using a tongue depressor and head light to improve visualization. (**B3**): Both palatine tonsils and the posterior oropharyngeal wall was swabbed. (**C1**,**C2**): The nasal swab was inserted approximately 1–3 cm into the nasal cavity and brushed along the septum and the inferior nasal concha. (**C3**): The swab was rotated three times and withdrawn.

**Figure 2 diagnostics-13-00283-f002:**
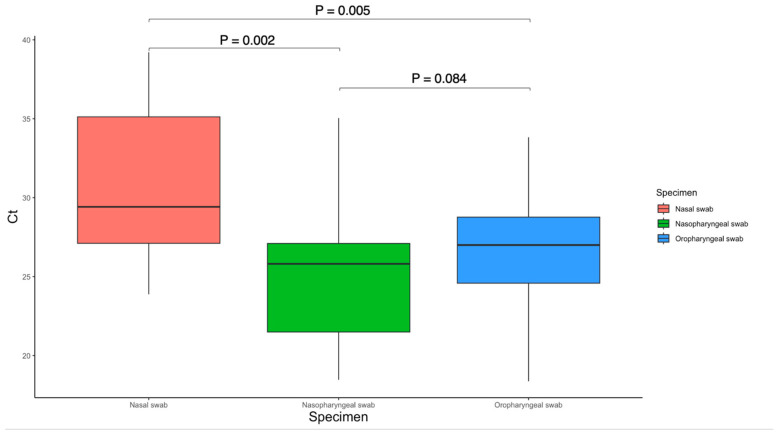
The Allplex^TM^ SARS-CoV-2 assay N gene target values (Ct) are illustrated in this box plot, N = 24.

**Table 1 diagnostics-13-00283-t001:** SARS-CoV-2 RT PCR results for oropharyngeal swab (OPS), nasopharyngeal swab (NPS), nasal swab and combined swabs. * 24 participants.

	*OPS*	*NPS*	*Nasal Swab*	*OPS/NPS*	*OPS/Nasal Swab*
*Positive*	48	47	42	51	49
*Negative*	3	4	9	0	2
*Sensitivity (%)*	94.1	92.2	82.4	100	96
*Mean Ct value*	26.63 *	24.98 *	30.60 *	-	-

**Table 2 diagnostics-13-00283-t002:** Comparisons of sensitivity between specimen types, N = 51.

Specimen Type	ΔSensitivity	*p*-Value *
OPS vs. NPS	1.6%	1.000
NPS vs. Nasal swab	9.8%	0.063
OPS vs. Nasal swab	11.7%	0.070
OPS/NPS vs. OPS	5.8%	0.250
OPS/NPS vs. NPS	7.8%	0.125
OPS/NPS vs. Nasal swab	17.6%	0.004
OPS/Nasal swab vs. OPS	3.9%	1.000
OPS/Nasal swab vs. NPS	5.8%	1.000
OPS/Nasal swab vs. NMTS	15.7%	0.031

OPS: oropharyngeal swab, NPS: nasopharyngeal swab, Δ% points difference in sensitivity. * McNemar test.

## Data Availability

Due to our nation’s strict data sharing policies, we are unable to make our data public. However, researchers who wish to access the data can contact the corresponding author, Kasper Daugaard Larsen. After permission has been granted by the regional data committee, data will be made available to the researchers.

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
