# Peer review of "Head-to-Head Comparison of Nasopharyngeal, Oropharyngeal and Nasal Swabs for SARS-CoV-2 Molecular Testing"

_diagnostics, 2023, doi:10.3390/diagnostics13020283_

Round 1

Reviewer 1 Report

General consideration

This study presents innovative diagnostic aspects related to the nucleic acid research of SARS-CoV-2. However, there are criticisms to correct before publication.

Major criticisms

- The number of subjects examined is small. Of these, 24 subjects were examined with a real-time PCR assay described in the Materials and Methods section, while the type of assay used was not reported for the remaining subjects examined elsewhere;

- the collection of the biological sample was not done on the same day for all the subjects examined and at a considerable distance from the detection of the first positive result;

- I suggest to the Authors to be more cautious in stating that ".. differences between systems should not impact..." (see page 6, lines 198-200), as the use of different methods with different sensitivities or targets differences could impact the detection of positive samples. In fact, a negative outcome could have turned out positive by another method;

- the ct-values reported in Table 1 were obtained for 24 participants and not for all, therefore it is improper to report this information without specifying it.

Minor criticisms

- The Materials and methods section does not indicate the number of participants in the various locations, the storage temperature of the sample and the time taken for transport to the laboratory.

Author Response

Thank you for your good and relevant comments to improve the manuscript. 
Please review our response below. 

Reviewer 2 Report

Dear Editor,

thank you for the opportunity to review the paper titled Head-to-head comparison of nasopharyngeal, oropharyngeal and nasal swabs for SARS-CoV-2 molecular testing. The manuscript is well-written and interesting, and the objectives are stated clearly. This is a prospective study and that is the strength of this study. Authors have stressed some limitations for instance that all swabbing were performed by otorhinolaryngologists and therefore the results should be carefully used in outpatient settings. 

However, there are issues that the authors need to address before the manuscript may be considered for publication. The following are my comments describing these issues.

The study has other limitations which are not clearly stated or discussed. The limitation of the study is the relatively small sample size. Then statistical analyses could also be underpowered and results should be interpreted with caution. It would be advisable to discuss these limitations. And then the conclusion should be in line with the mentioned limitations. The impact of variations in specimen collection on sensitivity is unknown, therefore, this could be mentioned in the conclusion and in the abstract too.  

And I would also advise authors to protect the identity of the person in figure 1.

Good luck!

Author Response

Thank you for your good and relevant comments to improve the manuscript. 

Round 2

Reviewer 1 Report

The article has been improved and the limitations noted.